# Identification of Novel lncRNA and Differentially Expressed Genes (DEGs) of Testicular Tissues among Cattle, Yak, and Cattle-Yak Associated with Male Infertility

**DOI:** 10.3390/ani11082420

**Published:** 2021-08-17

**Authors:** Shaokang Zhao, Tingting Chen, Xinmao Luo, Shiyi Chen, Jie Wang, Songjia Lai, Xianbo Jia

**Affiliations:** Farm Animal Genetic Resources Exploration and Innovation Key Laboratory of Sichuan Province, Sichuan Agricultural University, Chengdu 611130, China; z13503321824@163.com (S.Z.); s18884328298@163.com (T.C.); lxmrobert@163.com (X.L.); sychensau@gmail.com (S.C.); wjie68@163.com (J.W.); laisj5794@163.com (S.L.)

**Keywords:** cattle-yak, hybrid, lncRNA, DEGs, male sterility

## Abstract

**Simple Summary:**

Cattle-yak is an excellent hybrid of male cattle and female yak, which has many more outstanding production traits, such as better adaptability to high altitudes and better meat quality. However, the male sterility of cattle-yak restricts the utilization of superior heterosis. Few studies have focused on the comprehensive analysis of cattle-yak and its parents, in order to find factors on infertility of the cattle-yak. This study comprehensively analyzed the mRNA and lncRNA expression profiles of testicular tissue samples of cattle, yak, and cattle-yak by RNA-seq technology, and identified some differentially expressed genes that may be related to male sterility of cattle-yak, in order to provide a theoretical basis for solving the problem of breeding work.

**Abstract:**

Cattle-yak is an excellent hybrid of cattle and yak; they are characterized by better meat quality and stronger adaptability of harsh environments than their parents. However, male sterility of cattle-yak lay restraints on the transmission of heterosis. In this study, next generation sequence technology was performed to profile the testicular tissues transcriptome (lncRNA and mRNA) of cattle, yak, and cattle-yak. We analyzed the features and functions of significant differentially expressed genes among the three breeds. There are 9 DE lncRNAs and 46 DE mRNAs with comparisons of cattle, yak, and cattle-yak. Among them, the upregulated targeting genes, such as IGF1 and VGLL3 of cattle-yak lncRNA, may be related to the derangement of spermatocyte maturation and cell proliferation. Similarly, we found that the LDOC1 gene, which is related to the process of cellular apoptosis, is overexpressed in cattle-yak. GO enrichment analysis demonstrated that the cattle-yak is lacking the regulation of fertilization (GO: 0009566), spermatogenesis process (GO: 0007283), male gamete generation process (GO: 0048232), sexual reproduction (GO: 0019953), and multi-organism reproductive process (GO: 0044703), such processes may play important and positive roles in spermatogenesis and fertilization. Furthermore, the KEGG enrichment analysis showed that the upregulated DEGs of cattle-yak most enriched in Apoptosis (ko04210) and Hippo signaling pathway (ko04390), may lead to excessively dead of cell and inhibit cell growth, resulting in obstruction of meiosis and spermatogenesis processes. This study will enable us to deeper understand the mechanism of male cattle-yak infertility.

## 1. Introduction

Cattle-yak, as a hybrid progeny of male cattle (*Bos taurus*) and female yak (*Bos grunniens*), exhibit evident heterosis such as stronger adaptability of harsh environments, better meat quality and larger milk quantity production [1,2,3], which is of great significance to the economic development of the Qinghai Tibet Plateau [4]. However, because of male sterility (caused by spermatogenic arrest), heterosis cannot be inherited by the next generation, which limits the utilization of heterosis [2,5]. Therefore, the mechanisms of male sterility in cattle-yak need to be urgently investigated. Testicular histology revealed that gonocytes and spermatocytes of cattle-yak were established normally; however, spermatogenesis was arrested when the meiosis phase began at 10 months after the birth of hybrids, so the hybrid sperm producing function was handicapped, resulting in loss of sperm in semen [6,7]. Previous studies have proven that the chromosome morphology of the cattle-yak was different from its parents, and the incompatibility, due to deleterious interactions among genes of different species, was the main cause of hybrid unviability and sterility in most animals. For example, the male sterility of bovine hybrids, such as the F1 progeny of European bison and domestic cattle, was attributed to abnormal conjugation, resulting from a lack of sufficient homology between chromosomes from the parents, or due to disturbances in the divisional spindle formation in spermatocytes [8,9,10]. The abnormal meiotic process after 10 months may be caused by incompatibility of the parents’ chromosomes. Although the testis shape and size of seminiferous tubules can gradually return to normal levels of hybrids after F3 generation by backcross, the heterosis at this time is not very obvious. Therefore, the most important proposal is to resolve the sterility problem of F1–F3 generations.

With the development of molecular biology, some previous studies had preliminarily found the molecular mechanism of cattle-yak male sterility. As an important process of epigenetic regulation, precise histone modifications are essential for normal spermatogenesis. However, because of the abnormal regulation of histone methyltransferases (HMTS), the distributions of histone methylations of cattle-yak seemed unusual. For instance, immunofluorescence detection of selected histone methylations, in cross-sections of testicular tissues or meiotic chromosomes, demonstrated depletion of methylation histone 3 at lysine 4 (H3K4me3) and significant enrichment of methylation of histone 3 at lysine 27 (H3K27me3). The H3K4me3 is associated with the enhancer region and marks transcription activation while methylation of H3K27me3 labels the repressive state and gene silencing [11,12]. Furthermore, as the number of deleted azoospermia (DAZ) family, the bovine Boule (bBoule) gene promoter long CpG island (CGI) in the testes of cattle-yak will occur hypermethylation and lead to the low expression of the bBoule gene. This will lead to azoospermia and further infertility of cattle-yak [13,14]. The previous study compared the expression levels of Heat shock protein 27 (Hsp27) and protein 53 (P53) between cattle-yak and yak, as well as the expressions that differed across the testis developmental stages. The results showed a high expression level of P53 and a low expression level of Hsp27 in cattle-yak, and show the opposite in yak. Meanwhile, studies have shown that the inhibition of Hsp27 phosphorylation retarded P53 induction and resulted in cell apoptosis; the increase in HSPs and decrease in P53 could protect and restore the damaged reproductive system in the testis, so we can conclude that the upregulation of P53 would lead to cell apoptosis, abnormal meiosis, and further infertility [15,16]. Moreover, the downregulation of DMC1, DDX4, and DMRT7 involved in meiotic recombination and sexual development were also considered associated with spermatogenic arrest in cattle-yak [1]. A previous study identified 2960 genes differentially expressed between yak and cattle-yak with RNA-seq technology, in which several downregulated genes in cattle-yak were associated with cell cycle progression, meiosis, and sperm components [17,18]. However, few studies have focused on the male sterility of cattle-yak compared with cattle and yak.

In the current study, we analyzed the differences of gene expression in testicular tissues among cattle, yak, and cattle-yak by NGS technology and bioinformatics software. We screened for novel lncRNAs with the prediction of unknown transcripts and then identified the common differentially expressed lncRNAs and mRNAs in cattle, yak, and cattle-yak. With the function analysis of DEGs, we explored the relationship between male sterility of cattle-yak and gene expression levels. This study aims to clear the molecular mechanism of male sterility of cattle-yak and improve the breeding work in the future.

## 2. Materials and Methods

### 2.1. Ethical Statements

The study design was approved and all methods were performed in accordance with guidelines of the Institutional Animal Care and Use Committee at the College of Animal Science and Technology, Sichuan Agricultural University (permit number: DKY2020050). The minimal number of animals was used and special attention was paid to minimize animal suffering.

### 2.2. Sample Collection and Preparation

In order to ensure that the experimental results were statistically significant, we randomly selected three male cattle (H1, H2, and H3), three male cattle-yaks (P1, P2, and P3), and three male yaks (M1, M2, and M3) of consistent diet levels, with the same growth conditions, aged at 18 months, in a pasture in Hongyuan county, Sichuan, province of China. Then, we divided the nine samples into three groups (H, M, P). Testis samples of these animals were obtained when they were slaughtered. Testicular tissue samples were collected from each animal and stored at −80 °C until total RNA was extracted.

### 2.3. RNA Extraction, Library Preparation, and Illumina Sequencing

Total RNA was extracted from each sample with the TRIzol Reagent (Thermo Fisher Scientific, Shanghai, China) following the manufacturer’s instructions. The integrity of total RNA was detected with 1.8% agarose gel electrophoresis, respectively. Before construction of cDNA libraries, ribosomal RNA of the 9 total RNA samples were removed with the Ribo-off rRNA Depletion kit (Vazyme Biotech, Nanjing, China), then the strand-specific sequencing libraries were constructed with VAHTS™ Stranded mRNA-seq V2 Library Prep Kit for Illumina^®^ (Vazyme Biotech, Nanjing, China), according to the manufacturer’s instructions. Then, the libraries were tested with 8% polyacrylamide gel electrophoresis. The 300–500 bp cDNA fragments were extracted and the cDNA concentration was evaluated with Qubit2.0 (Life Technologies, Camarillo, CA, USA). The strand-specific sequencing libraries were finally sequenced on the Illumina sequencing platform (HiSeq X Ten). The paired-end raw reads in fastq format were generated with a base calling of CASAVA software after getting the original image data files from the Illumina platform [19,20,21].

### 2.4. Quality Control, Mapping, and Transcriptome Assembly

Clean data were obtained after filtering out reads with the adapter sequences, ploy-length less than 35 nt, and low quality reads with fastp [22]. All downstream analyses were based on the high quality clean data. The clean reads were then aligned to the reference genome (ARS-UCD1.2_Btau5.0.1Y.fa) and annotation (ARS-UCD1.2_Btau5.0.1Y.gtf) with HISAT2 [23]. Then, the transcriptome was assembled and merged together with StringTie based on the reads mapped to the reference genome [24].

### 2.5. Quantification of Gene Expression Levels

StringTie [25] (1.3.1) was used to calculate transcripts per million (TPM), which was the percent of FPKM. FPKM means fragments per kilobase of exon per million fragments mapped, calculated based on the length of the fragments and reads count mapped to this fragment. In this study, we used the TPM as the criteria of the gene expression level. It was calculated as follows:$$TPM_{i}=\frac{x_{i}}{L_{i}}*\frac{1}{\sum_{j}\frac{x_{j}}{L_{j}}}*10^{6}$$
$x_{i}=total\;exon\;fragment/reads$, $L_{i}=\frac{exon\;length}{\;kb}$.

### 2.6. Unknown Transcripts Prediction and LncRNA Analysis

GffCompare [26] software was used to predict the new transcription region from the result of the class code. The unknown transcripts were then used to screen for lncRNAs. The non-coding transcripts with lengths more than 200 nt and that had more than two exons were selected as lncRNA candidates, and further screened with CPC2 [27], CNCI [28], Pfam [29], and PLEK [30], to distinguish the protein-coding genes from the non-coding genes.

### 2.7. Target Genes Prediction and Differential Expression Analysis

In this study, we used two methods (cis and trans) to analyze the target genes of lncRNAs. For the cis-analysis, we predicted the target coding genes within 10KB upstream and downstream of lncRNA with bedtools; for the trans-analysis, the target genes were predicted by correlation analysis or co-expression analysis between lncRNA and protein coding genes. LncRNA is involved in many post-transcriptional processes. For example, some antisense lncRNAs may regulate gene silencing, transcription, and mRNA stability by binding to sense strand mRNA. Hence, we predicted the complementary binding between antisense lncRNA and mRNA with RNAplex, in order to comprehend the interaction between the non-coding RNA and mRNA more deeply.

For the samples without biological replicates, DE analysis of each sequenced library was performed using the DEGseq. Briefly, the trimmed mean of M-values (TMM) method of edge R was used to standardize the reads count, and then the DEGseq R package was used to analyze the difference between samples. For the samples with biological replicates, we used the DEseq2 R package based on a model using the negative binomial distribution. The resulting *p*-values were adjusted using the Benjamin and Hochberg’s approach to control the false discovery rate. Genes with an adjusted q-value < 0.05 and |Fold Change| > 2 were assigned as DE.

### 2.8. Functional Enrichment Analysis

Gene Ontology (GO) enrichment analysis was used to classify the lncRNA target genes and DEGs based on the specific biological functions, which could help us identify their functions and the relationship between genes and phenotypes. Kyoto Encyclopedia of Genes and Genomes (KEGG) pathway enrichment was used to identify some molecular pathways related to some characters, further revealing the molecular mechanism of biological phenomenon.

### 2.9. qPCR Validation and Statistical Analysis

To validate the accuracy of gene expression data from RNA-seq, we randomly selected 2 lncRNAs, 2 mRNAs, and 2 target genes of differently expressed among cattle, yak, and cattle-yak to perform quantitative reverse transcriptase PCR (qRT-PCR) analysis. The remaining total RNA after RNA-seq was used to prepare the cDNA with reverse transcription kit (Thermo Fisher Scientific, Shanghai, China), according to the manufacturer’s instructions. A SYBR Green assay was used to quantify the expression of each gene with the internal control GAPDH and RPS18 [31]. The 2^−ΔΔCT^ method was used to calculate the relative expression levels of DEGs. Student’s *t*-test with Graph Pad Prism 6 software was used to compare the differences among cattle, yak, and cattle-yak samples,* *p*-value < 0.05 was considered statistically significant and ** *p*-value < 0.01 was considered as extremely statistically significant.

## 3. Result

### 3.1. Sequencing and Reads Mapping

The cDNA libraries were generated using the Total RNA from nine testicular tissue samples of cattle, yak, and cattle-yak (H1, H2, H3, M1, M2, M3, P1, P2, and P3). There were 313,647,777 raw reads generated with RNA-seq and 309,850,056 clean reads generated with filtering adaptor sequences and low quality reads. The GC content of the reads of these nine samples was about 50%. Sequence comparison of clean reads of each sample with the designated reference genome was performed, and the efficiency of the comparison ranged from 88.14% to 94.47%. All Q30 values of the reads exceeded 94.67%, and the multiple mapped reads ratio was less than 3.15% (Table 1). Sequence comparison and subsequent analysis were carried out using the specified genomic UCD1.2 as a reference. The reads of the unique mapped to the reference genome were used for further analysis.

### 3.2. Quantification of Gene Expression Levels

The TPM method was used to quantify the gene expression levels of each samples and differential biotype genes. As shown, TPM distribution of each sample in three groups was closed (Figure 1a), and the distribution of novel lncRNA proportion was the most abundant (Figure 1b).

### 3.3. Differential Analysis between Three Groups and Nine Samples

Three cattle (H1, H2, H3), three yaks (M1, M2, M3), and three cattle-yaks (P1, P2, P3) were divided into cattle group (H), yak group (M), and cattle-yak group (P), respectively. The differences of transcripts of three groups are demonstrated in the sample distance boxplot (Figure 2a) and the differences among nine samples are shown in the PCA 3D map (Figure 2b). As shown in Figure 2a, the distance of group H is far from group P, which indicates that there were significant differences in gene expression levels between the cattle and cattle-yak. We also find that the differences among the three samples of each group are not obvious (Figure 2b).

### 3.4. LncRNA Screening

GffCompare was used to find the new transcription regions with the merged results, the number of unknown transcripts was 10,893 (Table 2). Four kinds of analysis software (CPC2, CNCI, PLEK, and Pfam) were used to identify the lncRNA from noncoding transcripts and then 5130 lncRNAs were screened (Figure 3). The screening results of the four kinds of software are presented in the Appendix A.

### 3.5. Prediction of lncRNA Target Genes and Differential Expression (DE) Analysis

In this study, we used cis and trans models to explore the possible functions of DE lncRNAs and predict target genes of lncRNAs. The lncRNA antisense analysis results showed the relation between lncRNAs and mRNAs. For example, we predicted 37,197 target genes of these lncRNAs in cis model, and we found some target genes of lncRNAs, such as SOX, FAS, DAZAP2, and KMT5A, are related to testicular development. In the process of differential expression (DE) analysis, we set |Fold Change| > 2 and q-value < 0.05 as the screening criteria. We identified 1367 DE lncRNAs and 5212 DE mRNAs in H_vs_P, 26 DE lncRNAs and 127 DE mRNAs in M_vs_P, and their differential number is displayed by the MA plot (Figure 4a–d). The number of differential lncRNAs and mRNAs for each combination was plotted with a Venn diagram (Figure 4e,f); there were 9 DE lncRNAs and 46 mRNA were co-expressed in H_vs_P and M_vs_P (Appendix A).

### 3.6. Functional Enrichment Analysis of Gene Ontology (GO)

GO enrichment was used to annotate screened DEGs of cattle, yak, and cattle-yak. In this study, topGO was used to analyze the functional enrichment; we also drew the scatter diagrams of enrichment results. As shown, the target genes of upregulated lncRNA in cattle were enriched in the male gamete generation, fertilization, and spermatogenesis processes that positively regulate the motility of sperm (Figure 5a), indicating that a lack of regulation of these lncRNAs may lead to stagnant spermatogenesis and infertility. The upregulated mRNA of cattle enriched in some functional processes was associated with reproduction, such as sexual reproduction, gamete generation, spermatogenesis processes, and some cellular function processes, such as the intracellular part, the organelle function process, cellular component organization, or biogenesis and catalytic activity that related to protein production and meiosis (Figure 5b). Yak upregulated mRNAs were involved in processes cellular component organization or biogenesis and transcription factor activity, protein-binding processes that associated with cellular function processes. (Figure 5c). All of the above results suggest that the male sterility of cattle-yak may be caused by lack of regulation of these processes. The detailed descriptions of the GO terms of cattle and yak upregulated lncRNA and mRNA target genes are available in Appendix A.

### 3.7. KEGG Enrichment Analysis

In order to clear the function of DEGs, we used the Kyoto Encyclopedia of Genes and Genomes (KEGG) database with scatter to annotate the enriched pathways. The upregulated lncRNA target genes of cattle were most enriched in the ubiquitin mediated proteolysis (ko04210), HIF−1 signaling pathway (ko04066), focal adhesion (ko04510), and PI3K-AKT signaling pathway (ko04151) (Figure 6a). These pathways may engage in the processes of protein modification, cell proliferation, transcription, and translation. On the contrary, the upregulated lncRNA target genes of cattle-yak were most enriched in apoptosis (ko04210) and the Hippo signaling pathway (ko04390), which may negatively regulate the spermatogenic cells (Figure 6b). There were no significant different lncRNA target genes between yak and cattle-yak. The upregulated mRNAs of cattle and yak were most enriched in the mTOR signaling pathway (ko04150), ubiquitin-mediated proteolysis (ko04210), protein processing in endoplasmic reticulum (ko04141), and so on (Figure 6c,d). These pathways play crucial roles in regulating cellular functions in many aspects. The pathway enrichment description of lncRNA and mRNA target genes of different samples are presented in Appendix A.

### 3.8. Validation of qRT-PCR

To confirm that the expression levels of DEGs were consistent with the levels shown in these sequencing analyses data, we randomly selected two lncRNA (XR_003032659.1, XR_003033672.1), two mRNAs (NM_001038065.1, NM_001040492.2), and two target genes (SOX10,BAP1) to compare the expression patterns of DEGs by qPCR (Figure 7) with the sequencing results. The comparisons showed that they were consistent.

## 4. Discussion

Crossbreeding is used to improve the performance of animals, especially under harsh conditions [32]. However, the sterility of hybrid offspring restricts the transmission of heterosis. This global issue, which has received much attention in recent decades, needs to be solved. Many studies have focused on this problem, in regard to rice [33], mice [34], fish [35], bovine [7], etc. In this study, we used high-throughput sequencing methods with bioinformatics analysis tools to analyze the transcriptome differences among the testicular tissue of cattle, yak, and cattle-yak, and explore the causes of infertility of cattle-yak. We found that the transcriptome profile distance between the cattle-yak and its parents, especially cattle, is obvious (Figure 2). We hope to further understand the molecular mechanism of hybrid male infertility to assist future breeding work.

With the development of molecular biology, RNA-seq technology has become a convenient way to obtain large quantities of transcriptome data and analyze differential gene expressions on a genome-wide scale [36,37]. LncRNA was shown to play a crucial role in regulating animal sperm development [38], which has high correlation with expression of neighboring mRNA [39,40,41]. In this study, we applied RNA-seq technology to analyze the profiles of testicular tissues of cattle, yak, and cattle-yak. In the current study, GffCompare software was used to find new transcription regions, and 10,893 unknown transcription regions were predicted (Figure 3). We used four kinds of analysis software (CNCI, CPC2, PLEK, and Pfam) to screen lncRNA; 5130 lncRNAs were identified. Differential analysis demonstrated that 1367 lncRNA and 5212 mRNAs were signed as DE (Figure 4). The common DE identification with the Venn diagram among cattle, yak, and cattle-yak showed that there were nine lncRNAs and 46 mRNAs were differentially expressed. Among these differentially RNA, we found that there were 8 upregulated lncRNAs, 31 upregulated mRNAs, 1 downregulated lncRNA, and 15 downregulated mRNAs in cattle-yak among these DEGs (Appendix A). The target genes, such as insulin-like growth factor 1 (IGF1), VGLL3 of lncRNAs MSTRG.4843.1 and MSTRG.308.1, were upregulated in cattle-yak. Insulin/IGF1 was verified as important for mitochondrial biogenesis and fusion in gonadal steroidogenic cells of prepubertal males. However, the expression level of IGF1 was lower in sexually mature animals [42]. The upregulated MSTRG.4843.1 may make IGF1 overexpressed and disrupt the maturation of spermatogonium.VGLL3 was shown to negatively regulate Sertoli cell proliferation in testis and inhibit growth of pubertal testis of Atlantic salmon associated with the Hippo pathway [43]. Therefore, the upregulation of VGLL3 may hinder the growth and development of the testicles. However, the specific mechanism of action of the two lncRNAs on target genes needs to be further explored. In contrast, the target gene KIAA1211 (CRACD) of lncRNA MSTRG.18068.3 was downregulated in cattle-yak compared to cattle and yak. KIAA1211(CRACD) was the capping protein inhibiting regulator of actin, which demonstrated high expressed in the adult mouse testis [44]. CRACD played a crucial role in the process of meiosis and spermatogenesis; it can be the spermatogenesis biomarker of azoospermia phenotypes [45]. Therefore, the downregulation of CRACD of male cattle-yak may lead to their azoospermia and sterility. In summary, these lncRNAs may influence the fertility of cattle-yak by targeting their target genes.

LDOC1 is a gene encoding leucine-zipper protein, the overexpression of the gene-induced externalization of the cell membrane phosphatidylserine and further apoptosis. At the same time, the transcription factor, MZF-1, was shown to interact with LDOC1 and enhance the activity of LDOC1 for inducing apoptosis [46]. The sperm apoptosis of cattle-yak may be caused by the overexpression of the LDOC1 gene. However, the other upregulated genes, such as TGS1, NAE1, FGFR1, and SLC41A1, need to be further analyzed.

GO enrichment analysis showed that the target genes of lncRNA upregulated in cattle were mainly enriched in male gamete generation, fertilization, and spermatogenesis processes; this suggests that lack of these lncRNA regulations may downregulate the target genes and further cause spermatogenesis stagnation. Similarly, the pathways of upregulated mRNA of cattle and yak were mainly enriched in the process organelle part and protein synthesis, demonstrating that the downregulation of these genes might hinder the maturation of spermatocytes. KEGG enrichment analysis showed that the upregulated lncRNA of cattle target gene VHL were enriched in the HIF-1 signaling pathway and ubiquitin mediated proteolysis, both of which are related to protein ubiquitination modification. As a negative regulator of the p53 tumor suppressor protein, E3 ubiquitin-protein ligase Mdm2 can maintain the p53 protein at the basal level by regulating its ubiquitination and degradation by the 26s proteasome [47]. The sterility of cattle-yak may be caused by the upregulation of P53 [15]. The upregulated lncRNAs of cattle-yak target genes PTPN13 and COL4A3 (ATS2) were enriched in the apoptosis and Hippo signaling pathways that may lead to abnormal apoptosis or growth arrest of sperm cells.

## 5. Conclusions

In conclusion, we identified nine lncRNAs and 46 mRNAs differentially expressed among cattle, yak, and cattle-yak. Most of them were specifically involved in the regulation of spermatozoa maturation, cell proliferation, and protein modification processes. Furthermore, the pathway enrichment analysis revealed that many processes were related to fertilization, spermatogenesis, and growth arrest of sperm cells. These results may provide more support for future research on the molecular mechanisms governing male cattle-yak infertility.

## Figures and Tables

**Figure 1 animals-11-02420-f001:**
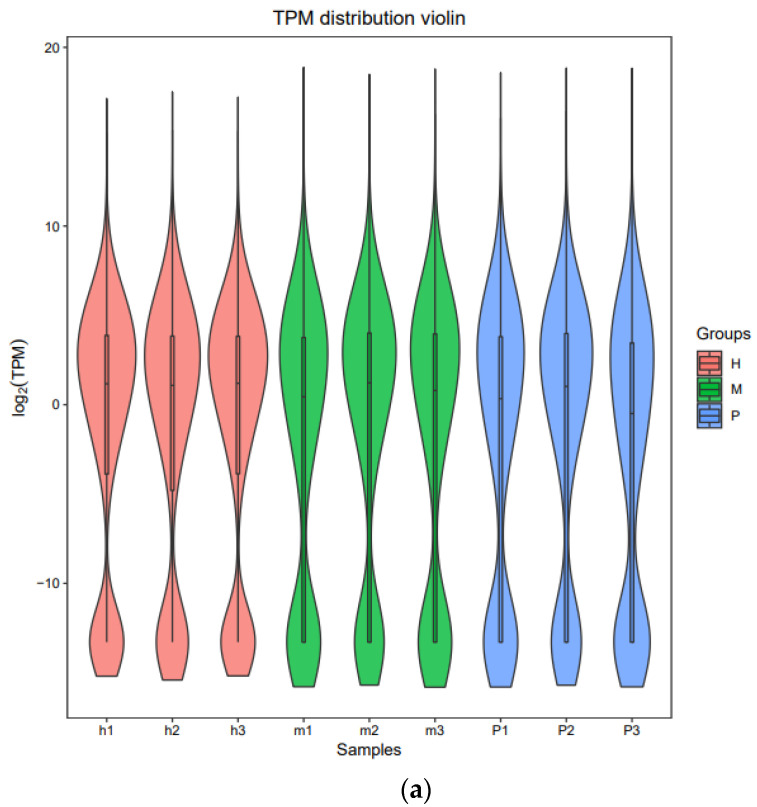
TPM distribution of nine samples in three groups and different RNA. (**a**) TPM distribution violin of nine samples (h1, h2, h3, m1, m2, m3, p1, p2, p3) in three groups (H, M, P) (**b**) TPM density distribution of Annotataed_lncRNA, novel_lncRNA, and mRNA.

**Figure 2 animals-11-02420-f002:**
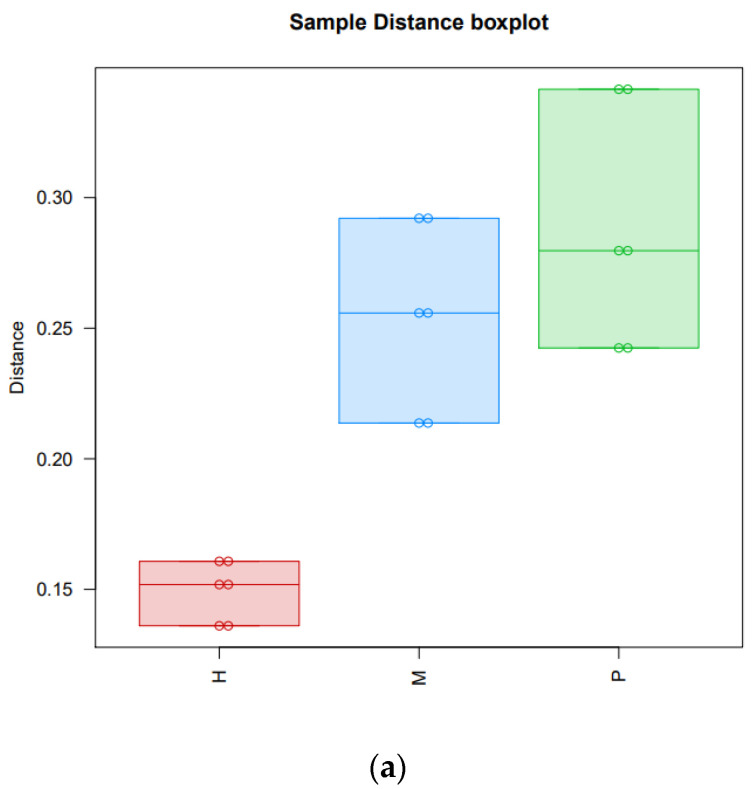
Differences among the three groups and nine samples. (**a**) The distance among three groups (H, M, P) with sample distance boxplot. (**b**) The distance among nine samples (h1, h2, h3, m1, m2, m3, p1, p2, p3) with the PCA 3D map.

**Figure 3 animals-11-02420-f003:**
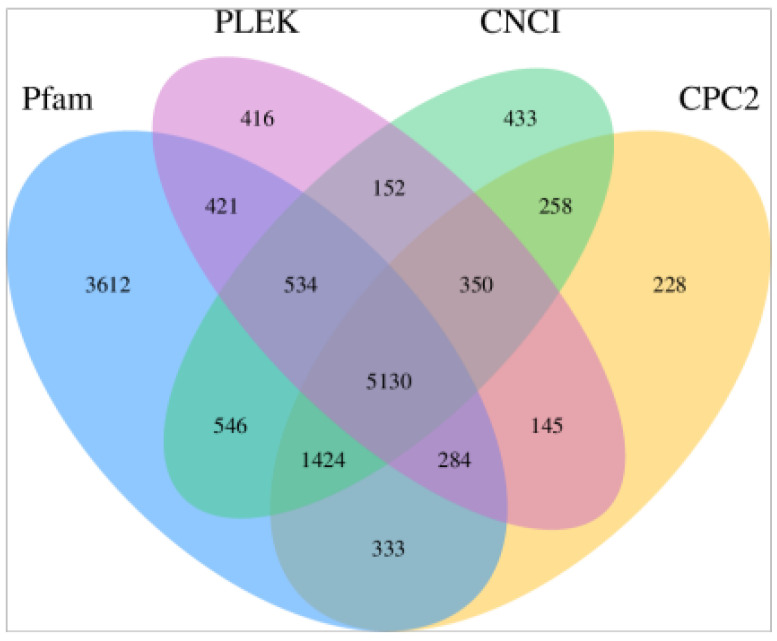
Venn diagram of lncRNA prediction of four tools (CNCI, CPC2, PLEK, and Pfam).The data shared by the four tools were designated as candidates for subsequent analysis.

**Figure 4 animals-11-02420-f004:**
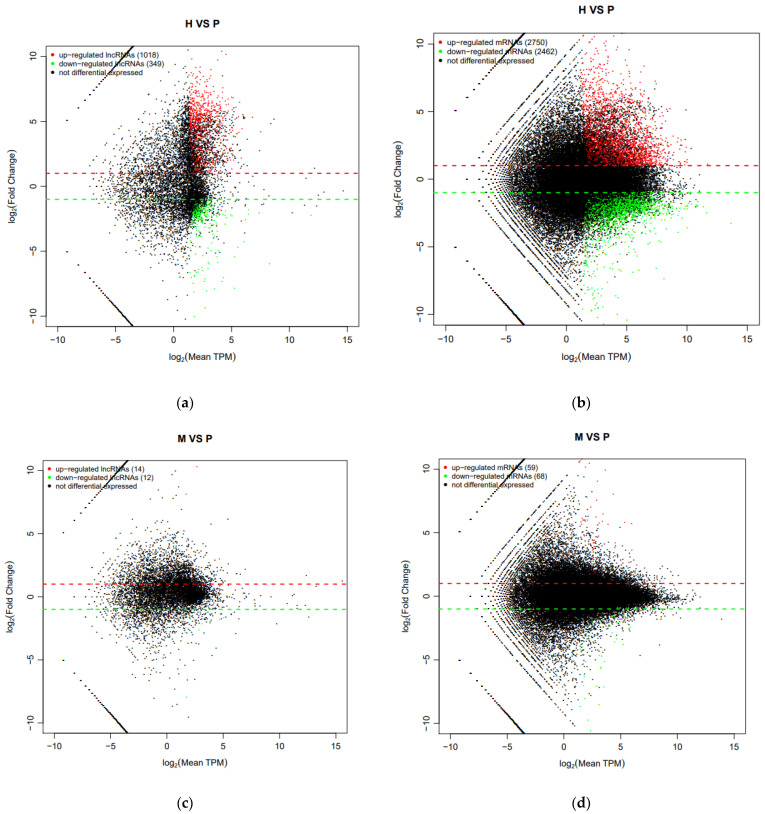
Differential expressed lncRNAs and mRNAs among cattle (H), yak (M), and cattle-yak (P). (**a**) MA plot of DE lncRNAs between H and P. (**b**) MA plot of DE mRNAs between H and P. (**c**) MA plot of DE lncRNAs between M and P. (**d**) MA plot of DE mRNAs between M and P. (**e**) Venn diagram of differentially expressed lncRNA between H_vs_P and M_vs_P. (**f**) Venn diagram of differentially expressed mRNA between H_vs_P and M_vs_P.

**Figure 5 animals-11-02420-f005:**
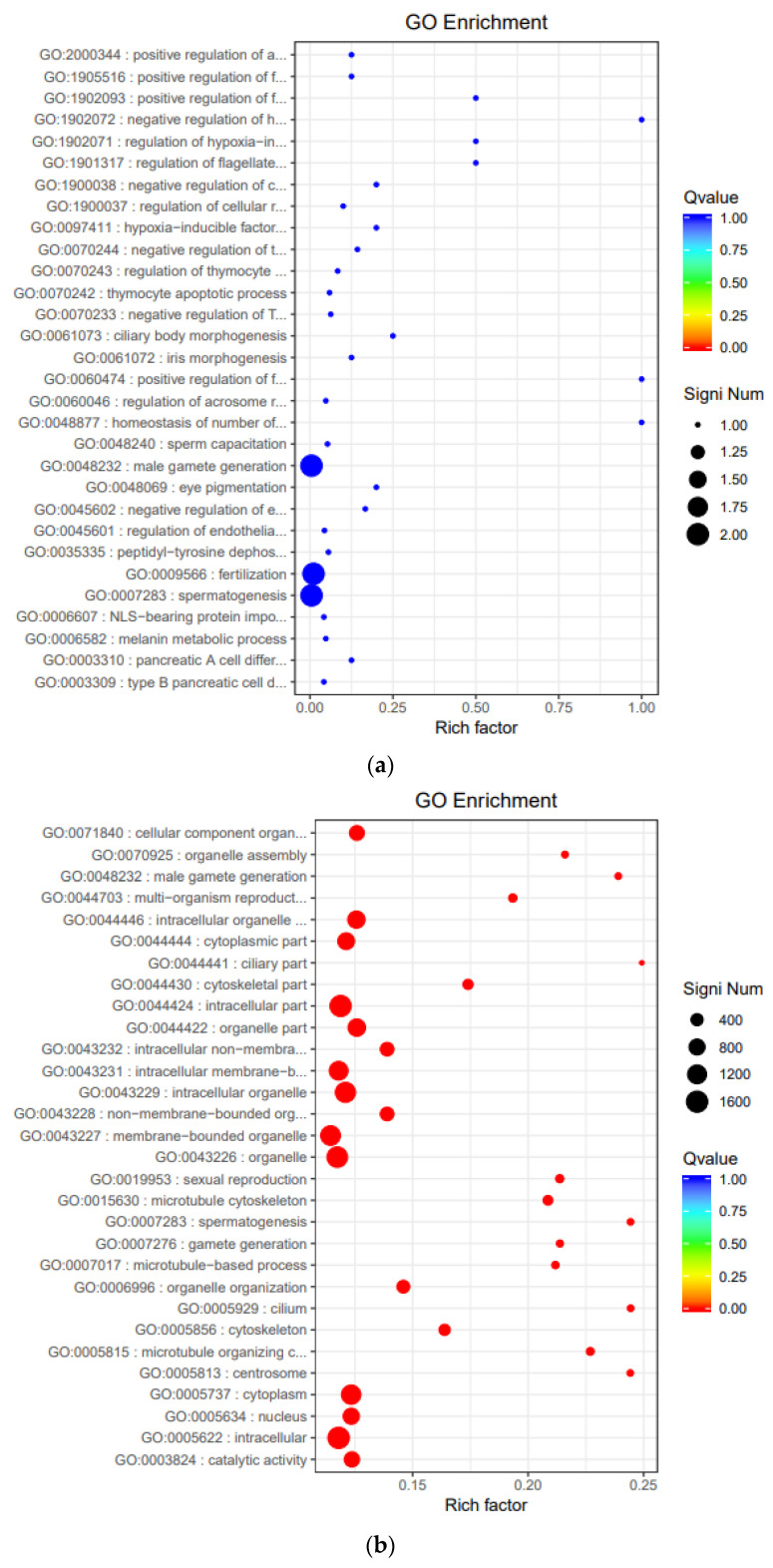
Scatter of GO distribution of target genes of LncRNAs and mRNAs differentially expressed among the three groups. (**a**) Scatter of GO distribution of upregulated lncRNA target genes of cattle. (**b**) Scatter of GO enrichment of upregulated mRNA genes of cattle. (**c**) Scatter of GO enrichment of upregulated mRNA genes of yak.

**Figure 6 animals-11-02420-f006:**
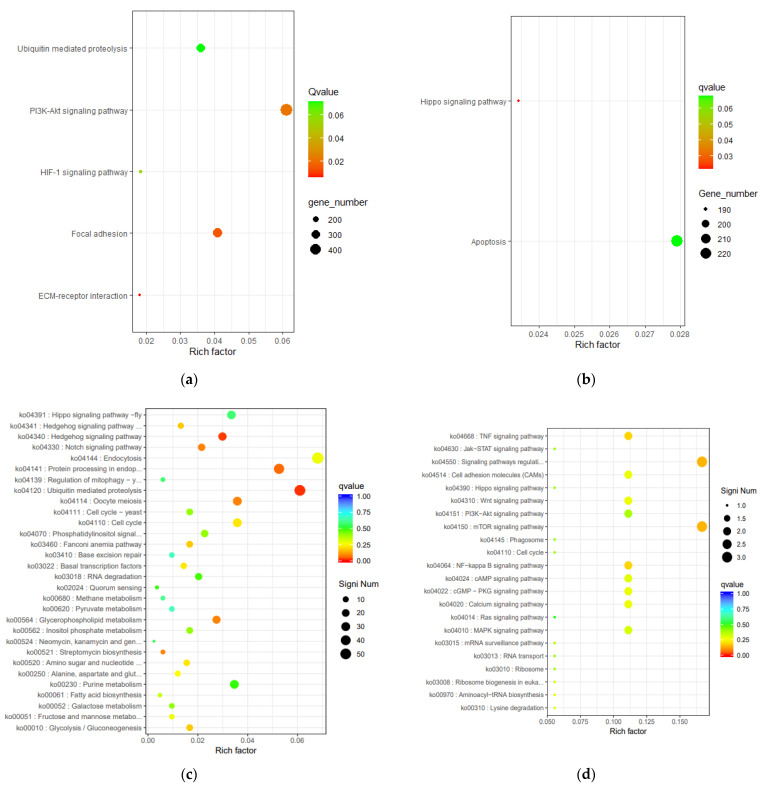
Scatter of KEGG enrichment of DEGs. (**a**) Scatter of KEGG pathways that the upregulated lncRNA target genes of cattle. (**b**) Scatter of KEGG pathways that the upregulated lncRNA target genes of cattle-yak. (**c**) Scatter of KEGG pathways that the upregulated mRNAs target genes of cattle. (**d**) Scatter of KEGG pathways that the upregulated mRNAs target genes of yak.

**Figure 7 animals-11-02420-f007:**
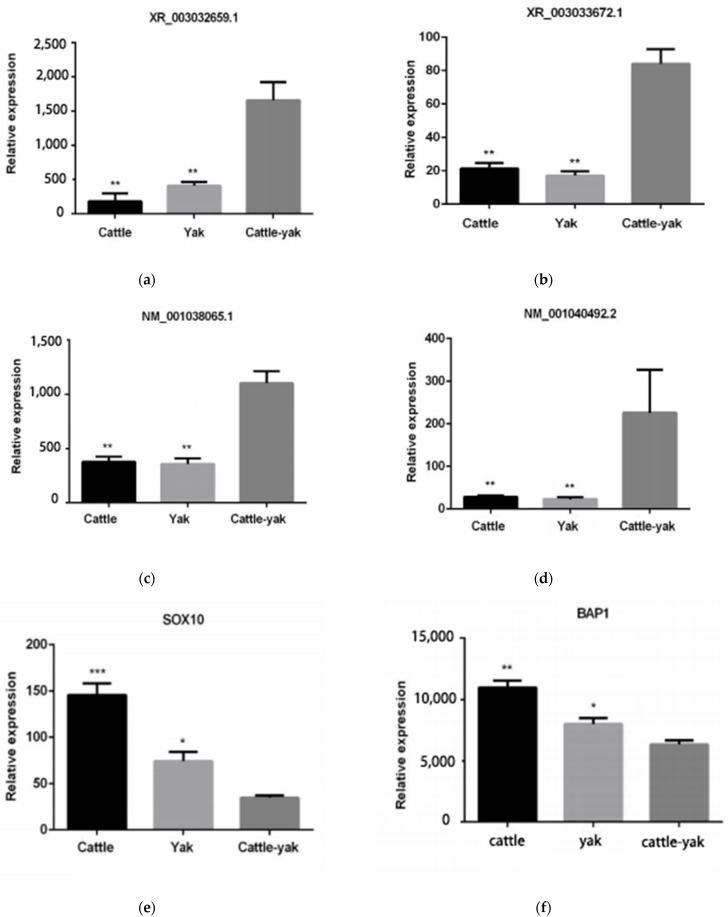
Validation of the differentially expressed genes by reverse transcription-real-time quantitative polymerase chain reaction (RT-qPCR). (**a**) The lncRNA XR_003032659.1 relative expression levels of cattle, yak, and cattle-yak. (**b**) The lncRNA XR_003033672.1 relative expression levels of cattle, yak, and cattle-yak. (**c**) The mRNA NM_001038065.1 relative expression levels of cattle, yak, and cattle-yak. (**d**) The mRNA NM_001040492.2 relative expression levels of cattle, yak, and cattle-yak. (**e**) The target gene SOX10 relative expression levels of cattle, yak, and cattle-yak. (**f**) The target gene BAP1 relative expression levels of cattle, yak, and cattle-yak. (* represents *p*-value < 0.05, ** represents *p*-value < 0.01).

**Table 1 animals-11-02420-t001:** Mapping results of the transcriptome data.

Samples	Total Reads	Total Map	Unique Map	Multiple Map	Q30 (%)	GC Content (%)
H1	66,835,092	62,423,261	59,139,082 (88.49%)	1,561,762 (2.34%)	95.14%	50.09%
H2	66,125,724	61,820,573	58,508,258 (88.48%)	1,627,394 (2.46%)	95.37%	50.19%
H3	72,510,220	67,528,071	63,952,926 (88.20%)	1,727,552 (2.38%)	95.18%	50.49%
M1	72,889,560	64,684,443	60,624,382 (83.17%)	1,808,268 (2.48%)	95.02%	50.45%
M2	67,418,572	59,919,256	55,651,456 (82.55%)	1,701,518 (2.52%)	95.17%	49.84%
M3	55,624,916	48,314,822	43,426,696 (78.07%)	1,083,154 (1.95%)	95.02%	50.57%
P1	68,746,398	60,688,223	56,257,718 (81.83%)	1,662,678 (2.42%)	95.04%	51.28%
P2	71,810,310	64,010,649	59,921,584 (83.42%)	1,673,008 (2.33%)	95.13%	50.64%
P3	85,334,762	76,830,344	71,423,888 (83.70%)	2,686,240 (3.15%)	94.67%	51.22%

**Table 2 animals-11-02420-t002:** The transcripts numbers of classification code with GffCompare.

Class Code	Class Code Description	Number
=	Complete match of intron chain	20,039
C	Contained	351
J	Potentially novel isoform (fragment): at least one splice junction	16,054
E	Single exon transfrag overlapping a reference exon and 10 bp intron	101
I	A transfrag falling entirely within a reference intron	6701
O	Generic exonic overlap with a reference transcript	845
R	Repeat. Currently determined by looking at the soft-masked	0
U	Unknown, intergenic transcript	10,893
P	Possible polymerase run-on fragment	725
X	Exonic overlap with reference on the opposite strand	3042
S	An intron of the transfrag overlaps reference intron	46
Y	No exon overlap: ref exons falling within transfrag introns	646

## Data Availability

All data generated or analyzed during this study are available from the corresponding authors.

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
