# Peer review of "Identification of Novel lncRNA and Differentially Expressed Genes (DEGs) of Testicular Tissues among Cattle, Yak, and Cattle-Yak Associated with Male Infertility"

_animals, 2021, doi:10.3390/ani11082420_

Round 1
Reviewer 1 Report
Overall:
This manuscript details an investigation into the male infertility of F1-F3 crosses of cattle and yak. This is an interesting concept and a worthwhile objective. However, there are several major considerations with this manuscript that need to be addressed.
First, the English language is poorly written and needs extensive editing. In addition, there are problems throughout the manuscript with spacing, in appropriate word choice, and statements that are not appropriate for scientific writing.
Second, the authors need to spend more time in the introduction explaining the problem. Infertility in hybrids is typically related to meiotic problems due to differences in structure or number of chromosomes. It does not appear that this is the case here but this needs to be established to establish a rationale for this study.
Lastly, additional detail and references are needed specifically in the methods section.
Specific comments:
Line 27-31- The last part of the abstract doesn’t make sense and needs revision
Line 36- Not sure what is meant here in relation to an evident heterosis of its parents
Line 88-89- An explanation is needed for the sample size of three in each group- this is very small and no power calculation is documented.
Line 125 – reference needed for Gffccompare software
Line 184-188 – this section is unclear and metrics to show effectiveness of sample size are needed.
Line 209-222- The GO term classifications seem to be interspersed with actual ontology terms and that makes this section very hard to interpret. For example cellular component organization being discussed with cell proliferation is confusing.
Tables- Table should be able to stand alone- Table titles are inconsistent and not descriptive enough.
Table 2- Meaning is not the correct title for the second column- I would suggest description and within this column transfag is not a complete word and should be better explained. In addition more information needs to be included about the column labeled number
Figure 1 and 2- Text in figures is very small and difficult to read
Figure descriptions should be expanded
Figure 3 Description is insufficient and text in figures is too small to read.
Figure 4-6 same as above
Figure 7- Box plots are fuzzy and need to be created at higher resolution for publication
Discussion- Discussion needs further detail to put results in context of current body of knowledge.
Reviewer 2 Report
The proposed manuscript needs minor corrections.
Spaces are missing in many positions (L 35,38,40,45, 60, …, 139, 147, … , 227, 228, 230, 234, …, 315, 318, 319, …, 373)
Line 66; “downregulation” → “down-regulation”
Lines 164-165; “*P value <0.05 164 was considered statistically significant and **P value <0.01 was considered as extremely 165 statistically significant.” → “(*P<0.05 **; P <0.01)”.
Line 176; “… ratio was less than 3.15%. (Table 1).Sequence …” → “… ratio was less than 3.15% (Table 1). Sequence …”
Line 193; “… screened(Fig.3) .The screening …” → “… screened (Figure 3). The screening …”
Line 216; “… that …” → “… that …”
Line 260; the resolution in Figure 3 should be better
Line 271; the text in Figure 7 should be more readable
Line 293; the text in Figure 6 should be more readable
Line 297; the resolution in Figure 7 should be better
Lines 312-313; delete the sentence “We hope to further understand the molecular mechanism of hybrids male infertility and help for the future breeding work.”
Lines 320-324; the sentence is part of the results (not part of the discussion) “In this study, we applied RNA-seq technology to analyze 320 the testicular tissue sequencing data of cattle, yak and cattle-yak, and then screened for 9 321 lncRNAs and 46 mRNAs were common differentially expressed and we found that there 322 were 8 up-regulated lncRNAs and 31 up-regulated mRNAs, 1 down-regulated lncRNA 323 and 15 down-regulated mRNAs in cattle-yak among these DEGs (Table S5).”
Lines; 380-386; write the initials of the name and surname instead of the full name and surname
Lines; 395-475; all references should be written according to journal instructions
Reviewer 3 Report
Review report
Line 9: There is no space after colon
Line 11: There is no space between “the” and “male”
Line 15: Please use “may be” instead just “may” or “may relate” instead “may related”
Line 17 as line 9
Line 18: There is space between “however” and comma
Line 24 as line 15
Line 26: There is no space after the dot
Line 26-30: Sentence unclear, please rephrase it
Line 30: Please use „give” instead „enable” or „enable us to understand”
Line 32: There is no space between “yak” and “hybrid”
Line 35: There is no space before parenthesis
Line 37: Please consider joining last sentence of this line with the previous one by “(…) [1-3], which is significant to the (…)”
Line 38: There is no space on both sides of parenthesis
Line 40: Did you mean “heterozygous”?
Line 40: as line 35
Line 41: as line 26
Line 43: The sentence is hard to understand. Does given period refers to meiosis or spermatogenesis arrest?
Line 44: Please consider phrase “after birth of hybrids”
Line 44 – 45: Please consider “sperm producing function was handicapped resulting in loss of sperm in semen”
Line 45: as line 35
Line 45: as line 26
Line 48: The sentence is hard to understand. Please rephrase it
Line 50: The introduction does not refer to chromosomal incompability of hybrids parents. It would be necessary to include a paragraph that refers to it, due to popular opinion that chromosomal incompability of parents is the main reason of progeny infertility and other molecular differences are secondary traits.
Line 52: Please use “modifications are”
Line 58: as line 35
Line 58: I do not understand that sentence. What is associated? The H3K4me3 with H3K27me3 or H3K4me3 with enhancer region? Please make that sentence more clear.
Line 60: There is no space before parenthesis and after the dot
Line 61: Please consider “had been verified as candidate genes”
Line 62: Is doubled “b” intended?
Line 63: as line 60
Line 63-66: The sentence does not fit the narration of the previous one. Please consider rephrasing it with beginning with P53 upregulation which can be the reason of infertility
Line 66: as line 35
Line 68: as line 60
Line 71: as line 35
Line 71: The sentence is hard to understand. Please rephrase it
Line 73: Please use “this” or “current”
Line 78: Please consider beginning with “”This study aims to clear”
Line 79 Please consider changing part of that sentence with “and improve the breeding in the future”
Line 84: There is no space after the colon and the dot
Line 88: Is the number of animals significant? There is stated that the animals were selected from pasture in one county. Is it possible for those animals to be related?
Line 95: Did you mean “Trizol”?
Line 96: as line 84
Line 97: as line 84
Line 104: as line 35
Line 104: Please use “PA”, the PAGE is an abbreviation of method name
Line 105: There are no detailed parameters for the sequencing. Please add them
Line 105: as line 60
Line 107: as line 35
Line 112: as line 35
Line 113: as line 35
Line 114: as line 35
Line 116: Please check if this is correct citation format for this journal
Line 125: Please fix the software name
Line 126: as line 26
Line 127: Please use “with” instead “have”
Line 129: Please consider “which can distinguish”
Line 133: as line 26
Line 136: as line 26
Line 138: as line 26
Line 139: There is no space after comma
Line 147: There is no space after “q-value”
Line 150-152: Please specify the reference organisms used for the analysis
Line 150: as line 35
Line 151: Please use “could” instead “can”
Line 152: as line 35
Line 154: Did you mean “revealing”?
Line 159: The validation should be performed on RNA originating from all organisms. What was the reason that only cattle-yak RNA was used? Please add validation for cattle and yak
Line 160: There is no provider of the transcription kit mentioned
Line 162: Why GAPDH and RPS18 were used as internal control genes? Is there any reference or was there preliminary research done that confirm that expression levels of that genes are constant in analysed tissue? Please compare: https://www.researchgate.net/publication/344212222_Identification_of_stable_internal_control_genes_for_accurate_normalization_of_real-time_quantitative_PCR_data_in_testicular_tissue_from_two_breeds_of_cattle
Line 163: as line 35
Line 163: If validation was performed only on RNA derived from cattle-yak, how the various RNAs were compared?
Line 175: as line 35
Line 176: as line 35
Line 176 Please consider moving table 1 into supplementaty files
Line 182: Is capitalisation of “novel” intended?
Line 182: Please move Figure 1 near the sentence, where it was mentioned
Line 186: Please move Figure 2 near the sentence, where it was mentioned
Line 190: Please correct the software name
Line 191 Please move Table 2 near the sentence, where it was mentioned
Line 193: Please move Figure 3 near the sentence, where it was mentioned
Line 194: Please consider phrase “are presented in” instead “were seen as”
Line 197: Please unify spaces before units and special characters (i.e. “>”) in the article
Line 197: Please unify the “q-value” spelling in the article
Line 199-200: Please move Figure 4 near the sentence, where it was mentioned
Line 202-207: Please consider removing that paragraph
Line 212: Did you mean “involved” instead “enriched”?
Line 213: Please move Figure 5 near the sentence, where it was mentioned
Line 213: as line 35
Line 215: as line 212
Line 216: Is the bold text intended?
Line 217: Please consider begining the sentence with :Yak up-regulated mRNAs were involved in processes”
Line 220: Did you mean “may be”?
Line 225: as line 35
Line226: as line 212
Line 227: as line 35
Line 228: as line 35
Line 228: Please move Figure 6 near the sentence, where it was mentioned
Line 230-232: as line 35
Line 232: as line 139
Line 234-235: Please correct missing spaces
Line 237: Please consider “are presented in the table S9-S12”
Line 242-243: as line 234
Line 239-243: Please consider moving that part into methods section
Line 246: Please unify name of that software in article
Line 260 Did you mean “Venn”?
Line 263: Is this sentence intended to be located in this line?
Line 293: Please consider “that up-regulate” instead “that up-regulated”
Line 303-363: Discussion needs major corrections and need to be written from the beginning. First part repeats or should be moved into introduction, whereas second part just concludes the results.
Line 365: Please consider removing “are common”
Line 368: Please consider removing “and so on”
Line 369: Please consider removing “etc”
Line 370: Please consider using “may” instead “will”
Figure 1: Please consider joining separate parts of that figure into one
Figure 2: Please consider joining separate parts of that figure into one. Why there was prepared 3D graph instead 2D. What value is presented on the third axis?
Figure 3: Is it possible to improve the figure resolution?
Figure 4: Please consider joining separate parts of that figure into one or two parts
Figure 5: Please consider joining separate parts of that figure into one
Figure 6: Please consider joining separate parts of that figure into one
Figure 7: Please consider joining separate parts of that figure into one or two parts. Is it possible to improve the figure resolution?
Author Response
- Thank you for reading our manuscript carefully and putting forward valuable suggestions for revision.We have carefully modified the manuscript according to your suggestions, including grammar, writing and punctuation. We have revised the manuscript with the"track changes" function of MS word and resubmitted our manuscript in the system .
- We have sorted out the explanations of some content questions in "word file",you can find it in attachment.

Round 2
Reviewer 1 Report
The authors have addressed all of my comments.
Reviewer 3 Report
The authors have responded to the comments and suggestions and the manuscript has been improved compared to the latest version. Nevertheless, some sentences are hard to follow and must be corrected due to the poor English language. The manuscript requires native speaker corrections.